# Cold-Blooded and on Purpose: A Review of the Biology of Proactive Aggression

**DOI:** 10.3390/brainsci11111412

**Published:** 2021-10-26

**Authors:** Kimberly D. Belfry, Nathan J. Kolla

**Affiliations:** 1Waypoint Research Institute, Waypoint Centre for Mental Health Care, Penetanguishene, ON L9M 1G3, Canada; kbelfry@waypointcentre.ca; 2Department of Psychiatry, University of Toronto, Toronto, ON M5S 1A1, Canada; 3Centre for Addiction and Mental Health (CAMH), Toronto, ON M5T 1R8, Canada; 4Violence Prevention Neurobiological Research Unit, CAMH, Toronto, ON M5T 1R8, Canada; 5Waypoint/University of Toronto Research Chair in Forensic Mental Health Science, Penetanguishene, ON L9M 1G3, Canada

**Keywords:** proactive aggression, autonomic function, HPA axis, twin studies, molecular genetics, event-related potentials, neuroimaging

## Abstract

Proactive aggression (PA) is a planned and unprovoked form of aggression that is most often enacted for personal gain or in anticipation of a reward. Frequently described as “cold-blooded” or goal oriented, PA is thought to be associated with low autonomic arousal. With this view in mind, we performed a scoping review of the biological correlates of PA and identified 74 relevant articles. Physiological findings indicated a robust association between PA and reduced resting heart rate, and to a lesser extent a relationship between PA and decreased heart rate and skin conductance reactivity, perhaps indicating dampened sympathetic function. The twin literature identified PA as a heritable trait, but little evidence implicates specific genes in the pathogenesis of PA. Neuroimaging studies of PA pinpoint impaired amygdala function in the assessment and conditioning of aversive stimuli, which may influence the establishment of behavioral patterns. Nodes of the default mode network were identified as possible neural correlates of PA, suggesting that altered function of this network may be involved in the genesis of PA. Given the overlap of PA with reactive aggression and the overall behavioral complexity of PA, it is clear that multiple endophenotypes of PA exist. This comprehensive review surveys the most salient neurobiologically informed research on PA.

## 1. Introduction

Aggression is defined as the noxious behavior of one individual directed at another person or object, in which verbal or physical force is used to injure, coerce, or express anger [1,2]. While the immediate consequences of aggressive acts may be limited to physical injury or damage to property, perpetual and maladaptive aggression comes at a substantial cost to society [3].

The bimodal distinction between reactive (RA) and proactive aggression (PA) is one of the most widely utilized aggression classification schemes [4,5,6]. RA is enacted in response to threatening or hostile stimuli and is thought to be associated with high arousal, impulsivity, and uncontrolled behavior [7,8]. In contrast, PA is collected, goal-oriented, and instrumental in nature [1,9]. In children, PA is notably related to callous–unemotional (CU) traits and bullying [10,11], whereas PA in adults is frequently observed in offending populations [12,13]. There is unquestionably behavioral overlap between RA and PA; nonetheless, significant evidence points to related yet separate endocrinological, genetic, and neurobiological mechanisms of the two aggression subtypes [14,15,16,17].

Aggression research to date has largely focused on reactive types of aggression and most studies that evaluate PA do so in tandem with RA. To the best of our knowledge, no review to date has evaluated the biological underpinnings of PA. In an attempt to address this gap, we conducted a scoping review of the extant literature on the biological correlates of PA in humans. The aims of this review were the following: (1) to algorithmically identify all studies in the extant literature that explicitly link PA to some aspect of neurobiology, including brain structure and function; (2) to group the identified articles according to area of study and critically review and contrast research findings; and (3) to identify significant gaps in the literature to guide future research.

## 2. Measures of Proactive Aggression

There are several validated tools used to quantify PA. In general, the available methods can be classified according to the following three categories: (1) survey-based assessments, (2) laboratory-based paradigms, and (3) using historical behaviors as a proxy for PA. Survey-based evaluations are relatively inexpensive, simple to administer, and are the most commonly used, followed by aggression paradigms and historical behaviors. However, recognizing the corresponding strengths and limitations of the various tools is critical when selecting the most appropriate methodology.

Dodge and Coie [4] were the first to validate a survey-based measure of PA and RA. They developed the Teacher Rating Instrument (TRI), a 24-item questionnaire that uses two three-item subscales to assess PA and RA. This instrument has had widespread use and has been the basis for subsequent instruments measuring PA, namely, the Reactive–Proactive Questionnaire (RPQ) [11]. The RPQ is the most readily utilized tool to quantify PA to date. It has high internal consistency and discriminant validity, is generalizable across various cultural demographics, and has been validated for use in child, adolescent, and adult samples [18,19]. Specialized instruments, such as the Self-Report of Aggression and Social Behavior Measure (SRASBM) [20], can be used to measure relational PA, which involves attempting to harm others by threatening or damaging interpersonal relationships [21,22], whereas the Children’s Scale of Hostility of Aggression: Reactive/Proactive (C-SHARP) is designed for administration on children with developmental disabilities [23]. A common limitation of survey-based instruments is rater bias [24]. For example, one may be inclined to minimize or embellish when self-reporting, while teacher reports may be influenced by perceiving the subject as a “good” or “poor” student. Furthermore, survey assessments are limited in their ability to explore causal relationships between variables and generally focus on behavioral patterns (e.g., trait PA) and not in-the-moment behavior (e.g., state PA) [25].

Aggression paradigms are laboratory-based measures that employ simulated situations to elicit quantifiable aggressive behavior in the subject. Typically, the subject is instructed to compete against a fictitious opponent and is incentivized with points or a monetary reward. The subject may aggress the opponent by interfering with, stealing from, or by administering punishments (e.g., electrical shocks or loud noise blasts), where PA is measured as the intensity and number of aggressions the subject directs at the opponent in the absence of provocation. Commonly used paradigms include the Taylor Aggression Paradigm (TAP) [26] and the Point Subtraction Aggression Paradigm (PSAP) [27]. Paradigms are limited in that they reflect aggression that is artificially sanctioned by a third party, which is generally low in intensity [28,29].

Lastly, PA may be inferred by using past behavior as a proxy. Data are most often obtained from offending records, which allows subjects to be categorized as having engaged in PA or not. This approach theoretically measures organic PA that is not influenced by data collection methods. However, it is difficult to label aggressive behavior as purely proactive in nature, especially when data are gleaned only from records. This methodology also assumes that past behavior is predictive of current and future behavior. Past aggression remains the most consistent and stable predictor of future aggression [30]. PA scores have accordingly shown to be predictive of delinquency and conduct problems in boys [31,32] and violent criminal recidivism among adults [33], much of which has been shown to be proactive in nature.

## 3. Article Screening

To inform our review, the following keywords, in various combinations, were entered into MEDLINE/PubMed, PsychINFO, and EMBASE databases on May 2020: proactive, premeditated, predatory, planned, unprovoked, offensive, instrumental, or cold-blooded aggression; AND genetic, twin, neuroimaging, neurotransmitter, physiological, or hormone. We restricted our search to articles published in English and excluded animal studies. We scanned systematic reviews and meta-analyses to locate additional articles that were not captured by our search terms.

Our search identified a total of 6113 articles (Figure 1). Of the initial search results, we removed 292 articles that were not published in English and 643 that were duplicates. Therefore, 5178 articles were screened at the title level, 4775 at the abstract level, and 646 at the full-text level. As per our eligibility criteria, 70 studies were included in our review and an additional four articles were identified by hand searching. The resultant 74 studies were categorized as follows: psychophysiology (22 articles), hormones (18 articles), twin studies (6 articles), molecular genetics (11 articles), neurophysiology (5 articles), and neuroimaging (14 articles). Our most pertinent findings have been summarized in Table 1, Table 2, Table 3, Table 4, Table 5 and Table 6.

## 4. Psychophysiology

The psychophysiology of aggression measures the biological expression of mental processes in an attempt to identify differences that may be characteristic of aggressive behavior. The so-called “fight or flight” response is controlled by the sympathetic and parasympathetic branches of the autonomic nervous system (ANS) in order to regulate critical life function [114]. The sympathetic nervous system (SNS) is responsible for preparing the body for situations of threat or danger and its arousal is most frequently quantified in the laboratory by heart rate (HR) or by skin conductance (SC), which is a measure of stress-related perspiration [115]. Assessments of resting skin conductance (RSC) and resting heart rate (RHR) reflect one’s innate or baseline SNS arousal, whereas skin conductance reactivity (SCR) and heat rate reactivity (HRR) are a measure of sympathetic function under stress (e.g., [35]).

The parasympathetic nervous system (PNS) controls the conservation of energy and functions to return the body to equilibrium. Heart rate variability (HRV) and respiratory sinus arrhythmia (RSA) are two parameters commonly used to assess PNS activity. HRV is the beat-to-beat variation in the heart and RSA refers to transient increases and decreases in HR that correspond to inspirations and expirations due to the influence of the vagus nerve on the sinoatrial node of the heart [116,117,118]. Both HRV and RSA reflect adaptation to change, where increased variability indicates heightened ANS balance [119,120]. Vagal tone, as indexed by RSA, may be evaluated at rest (RRSA) or under conditions of reactivity (RSAR). Higher vagal tone has been hypothesized to relate to enhanced emotional regulation [115,121], whereas reduced vagal tone has been linked to behavioral problems, including aggression [118].

It has been suggested that autonomic under-arousal may underlie PA [122]. Two possible explanations have been proposed for this association: (1) chronically under-aroused individuals seek out risky situations to increase their arousal level to a more desirable state [123,124]; and (2) under-arousal reflects fearlessness [125], which in turn impairs one’s ability to learn from punishment, a primary mechanism of socialization [122,126]. In either case, a pattern of disinhibition may predispose under-aroused individuals to assign greater salience to rewards or the achievement of goals [127]. This is in contrast to RA, which is associated with heightened emotional and physiological arousal thought to reflect an automatic stress response [122]. The study of psychophysiology as it relates to aggression is important to our understanding of how individual differences in ANS function might correspond to behavioral problems or the emergence of criminal behaviors. Child and adolescent cohorts are especially important, as they represent demographic populations that are more malleable and theoretically more likely to benefit from targeted interventions intended to mitigate life-course maladaptive behaviors.

### 4.1. Studies of Children

Xu and colleagues [41] studied Chinese children at grades two and four and reported a negative association between aggression (PA and RA) and RHR. They suggested that low RHR may be predictive of aggressive preparedness, irrespective of aggressive subtype. Another study of Chinese children likewise reported dampened RHR for both PA and RA. The relationship between PA and RHR remained significant after controlling for RA, but not vice versa, indicating a stronger association between PA and SNS activity at rest. This relationship has also been reported in European boys with conduct disorder or oppositional defiance disorder (CD/ODD) [45]. Using the Instrument for Reactive and Proactive Aggression (IRPA) [24], the authors identified PA as a negative predictor of RHR, HRR, and SCR and a positive predictor of HRV. These results suggest that PA is associated with overall damped SNS activity (during rest and stress) but greater PNS activity when under stress. Others have identified dampened SNS stress responses for PA among North American child cohorts [15,49]. Using a paradigm task that had the subjects create art which could then be competitively criticized or destroyed by a fictitious opponent, PA was found to be negatively associated with SCR and RHR [15,49] but positively associated with RSAR [49]. These paradigm findings point to an overall reduction in physiological arousal for PA.

In an attempt to disentangle the autonomic differences between childhood RA and PA, Scarpa and coauthors [35] evaluated the relationships between aggression (PA and RA) and RHR, HRV, and SCR in a sample of North American children. PA was positively related to HRV and RSC and RA was negatively associated with HRV and RSC. Since these results could provide only partial support for the theory that autonomic under-arousal underlies PA [122], the authors concluded that PA and RA have unique physiological profiles with distinctive pathways to the manifestation of aggression. They further proposed that heightened RSC may reflect frontal lobe activation for individuals engaging in PA, as increased frontal activity, measured by magnetic resonance imaging (MRI), has been previously linked to upregulated SC [128,129]. In other words, heightened RSC may be indicative of improved executive function and the ability to delay gratification when in pursuit of a goal, which is required for PA [130,131]. Others have reported non-significant associations between PA, SCR, and RRSA in North American children [48]. The authors suggested that PA may not be characterized solely by depressed ANS function but instead by biosocial models that examine both biological (e.g., ANS and PNS function) and environmental risk factors (e.g., inconsistent discipline). Table 1 reflects all findings on psychophysiology and PA in child cohorts.

### 4.2. Adolescent Findings

Among adolescent studies, Crozier and coauthors [34] investigated the cardiac predictors of antisocial behavior in a large, North American, mixed-sex sample (*n* = 585). PA was negatively correlated with RHR in males, which may have been influenced by higher mean PA scores for males than females. The researchers hypothesized that low RHR is driven by temperament, whereas HRR is related to life experiences, such as early physical abuse or chronic social rejection. Relatedly, others have reported negative associations between PA and RHR and PA and HRR among adolescent boys [39]. Statistical analyses indicated that impulsive sensation seeking significantly mediated the relationship between PA and low RHR, thus inferring that chronically under-aroused individuals will seek out stimulating behaviors to raise their arousal to a more desirable level. These findings were not supported by Muñoz Centifanti and colleagues [38], who reported no significant associations between PA and HRR or SCR among detained boys. It is possible that this discrepancy may have been related to the use of the Competitive Reaction Time Task (CRTT) [57]. The CRTT measures preemptive aggression (e.g., aggression in the absence of provocation) as a proxy of PA, which may differ subtly from questionnaire-based methods (e.g., RPQ). Notably, there have been no significant reports of PA and psychophysiological findings assessed using the PSAP. Non-significant associations have also been reported in adolescents between PA and RSA [38,50], which coincides with findings from child cohorts.

As under-arousal has been hypothesized to reflect fearlessness [125], some have also suggested that deficient fear conditioning could be associated with PA [36,42]. Fear conditioning is a type of Pavlovian conditioning where an individual learns the aversive significance of a previously neutral stimuli through a process of association [132]. Using a sample of North American children, Gao and coauthors [42] longitudinally assessed PA at ages 10, 12, 15, and 18 years and collected fear-conditioned SC response data at age 18 years. Individuals who were persistently high in PA were significantly lower in conditioned SC responses. The authors hypothesized that being unable to learn from punishment may influence risky decision making and delay the development of conscience, which, in turn, manifests as persistent PA. Furthermore, impaired fear conditioning may be interpreted as indirect evidence of abnormal amygdala functioning [133,134], possibly indicating that deficiencies in learning from aversive stimuli may relate to individual differences in neural threat processing. The findings on psychophysiology and PA in adolescent cohorts are summarized in Table 1.

### 4.3. Adult Findings

Much of the adult psychophysiological literature on PA is based on healthy cohorts of university undergraduate students, which may lack generalizability to more antisocial populations. One such study compared PA, measured by the TAP, to cardiac function among male and female young adults and found PA to be related to dampened RHR but unrelated to HRV [54]. Alternatively, PA (measured by the RPQ) has been positively linked to RHR, but only in females, in a mixed-sex sample of university students [51]. This finding is largely inconsistent with the published literature on RHR and PA (Table 1); however, the authors cited that female PA scores were low and lacked variability. The researchers further noted a negative association between PA and SCR that was specific to males, tentatively suggesting sex differences. Sex-specific differences have been similarly reported by Bobadilla and colleagues [36], who evaluated conditioned SC and PA (measured by the TAP). The sample of 122 maladaptive undergraduate students was mass screened out of a potential 5733 students using items selected from the Narcissistic Personality Inventory [135]. Regression analyses determined that SCR was a negative predictor of PA across the whole sample but that the association was significantly stronger for males. In both cases, the relationship between SCR and PA remained significant after controlling for RA. The authors concluded that deficits in sensitivity to anxiety or punishment, as indexed by SC conditioned responses, were relevant to the manifestation of PA.

Despite the hypothesis that upregulated RSA reflects improved emotion regulation [115,121], the existence of a direct relationship between PA and RSA is not supported by the adult literature. One study of ethnically diverse females determined that the association between RSAR and aggression was confined to more direct forms of RA (e.g., overt physical attack or verbal assault) [53]. It has been postulated that biological vulnerability to emotion dysregulation (RSA) may mediate the relationship between borderline personality disorder (BPD) symptomatology and PA [52]; however, RSA and PA were unrelated in this study. Zhang and Gao [44] similarly reported no direct association between PA and RSA in a mixed-sex sample but identified RSA as a predictor of PA at low levels of social adversity (e.g., mentally ill parent or single-parent families). The authors contended that this association supports the “social push” theory [136], whereby the links between biological risk and antisocial behavior are most salient when the influence of social adversity is low.

Several studies have evaluated the psychophysiology of relational PA, a covert form of aggression that involves attempting to proactively harm another individual by threatening or damaging their interpersonal relationships [21,22]. No significant associations between relational PA and RSAR, HRR, or SCR were identified (Table 1), which is somewhat inconsistent with the results reported thus far. However, non-significant associations between PA and RSAR among female [37] and mixed-sex [43,47] cohorts support the work of Hagan and colleagues [53], who posited that RSAR is related to more overt forms of aggression. Blunted “fight or flight” responses to stress have also been reported for relational aggression in adolescent girls [137], suggesting that physiological reactivity may differ according to the type of aggression. Nonetheless, these studies are based on healthy populations and may not translate to clinically aggressive samples. Findings from adult studies of psychophysiology and PA are reported in Table 1.

### 4.4. Psychophysiology Conclusions

Upon review of the PA physiology literature, we suggest that the majority of the results support the tenet that autonomic hypo-arousal underlies PA [122]. There are moderately strong associations for PA with dampened RHR and less strong associations with HRR and SCR. Alternatively, lower SCR is associated with deficient fear conditioning [36,42], possibly indicating that physiological responses specific to PA may influence, or be influenced by, the processing of aversive stimuli. These findings could help explain why individuals high in PA are less likely to be dissuaded by the negative consequences of their actions [138]. Non-significant associations were reported between relational PA and SCR or RHR, which points to a potentially unique physiological basis underlying relational PA.

Compared to SNS function, only a limited number of studies reported significant findings relating PA to PNS activity. These included positive associations between PA and HRV but only among child cohorts. Associations between PA and RSA (at rest or under stress) were largely non-significant. Therefore, the “cold-blooded” nature of PA may be uniquely attributable to dampened SNS function. Meta-analytic review could help resolve the heterogeneity of psychophysiological findings published to date.

## 5. Hormones

Endocrine and nervous systems frequently coordinate their responses in times of stress or provocation [139]. In contrast to autonomic responses that are mediated by instantaneous nerve impulses, hormonal changes are the result of a chemical cascade and are relatively slower in response. Hormones regulate myriad bodily processes and attempts to establish a clear link between hormonal responses and aggression to date have been less precise. In a review on hormones and aggression across childhood and adolescence, Ramirez [140] concluded that hormones can be involved in the manifestation of aggression as a cause, consequence, or mediator.

### 5.1. Cortisol

Often termed the stress hormone, cortisol is an adrenal hormone whose concentrations are tightly regulated by a group of hormone secreting glands called the hypothalamic pituitary adrenal (HPA) axis [141]. When confronted with a stressor, the HPA axis triggers a molecular cascade that ultimately releases cortisol into the bloodstream, resulting in the mobilization of energy stores to prepare the body for fight or flight. In addition to stress responses, cortisol is also secreted in a pulsatile manner that displays a circadian rhythm pattern [142]. Daily cortisol levels typically peak 90 min after waking and steadily decline throughout the day, a phenomenon known as the cortisol awakening response (CAR) [141]. As cortisol is secreted into the blood, it may be sampled directly from the plasma or indirectly in the saliva; the two measures are highly correlated and both are considered experimentally valid [143,144]. There is evidence that certain traits such as callousness, unemotionality, and antisociality are linked to HPA axis hypo-activity [145,146]. However, this relationship is not clear-cut, as several studies have also indicated a positive relationship between cortisol levels and aggression [62,64,147].

#### 5.1.1. Child Findings

Among typically developing children, peer-nominated PA has been negatively associated with afternoon salivary cortisol [67,71]. While this finding has been confirmed by regression analysis, both PA and RA have also been found to be associated with dysregulated anger, perhaps indicating that any relationship between aggression (PA or RA) and afternoon salivary cortisol level is more reactively oriented [71]. Both studies also assessed teacher-reported PA and found no significant associations with cortisol. PA has been found to be similarly unrelated to salivary cortisol in depressed and anxious children [68]. While negative correlational findings between PA and plasma cortisol have been reported among females with a history of psychiatric hospitalization, PA and RA scores were highly correlated [70]. Furthermore, quantification of plasma cortisol can be inherently challenging, as the sampling process itself can be anxiety- and stress-inducing.

With respect to cortisol reactivity, one European study of disruptive behavior disorder (DBD) children and control subjects examined the relationships between salivary cortisol responses, RA, and PA, measured operationally according to spontaneous and delayed responses during dyadic play sessions (e.g., fishing game, zoo game, free play, and dominoes) and by parent report [63]. Despite identifying no differences in cortisol levels between control and DBD groups, the researchers were able to categorize participants as “responders” or “non-responders”, depending on whether the subject’s cortisol level increased or remained constant during the play sessions. Play-session-rated and parent-rated PA were largely inconsistent, with similar results being found only for DBD non-responders. These results may mean that correspondence between parent-observed PA and operationally-measured PA depend on cortisol response; however, these findings are preliminary and require verification. Another group measured PA experimentally using a task designed to elicit either fear or frustration and reported that RA, but not PA, significantly predicted total and peak post-stress salivary cortisol [64]. Interestingly, this study also found that children with high PA scores showed neither variable nor blunted HPA axis stress responses and thus responded comparably to the non-aggressive children. The findings for cortisol and PA in child cohorts are reported in Table 2.

#### 5.1.2. Adolescent and Young Adult Findings

PA has been found to be unrelated to salivary cortisol among ODD/CD or autism spectrum disorder (ASD) subjects [73], whereas a positive correlation has been reported for plasma cortisol and PA in high-risk adolescent males who experienced perinatal insults and family adversity [66]. In a recent study, a positive association between salivary cortisol level and PA was reported for adolescents anticipating peer interaction [50]. The authors posited that cortisol levels in victims of bullying may be elevated when anticipating peer rejection and may indicate that PA–cortisol associations are better explained by biosocial models that consider social risk factors. Most cortisol and PA literature has focused on child and adolescent cohorts; however, one study of North American college students compared the CAR to relational and physical subtypes of PA and RA using the SRASBM [69]. The only significant finding to emerge was a negative association between physical RA and CAR. The relationships between cortisol and PA in adolescents and adults that have been reviewed are reflected in Table 2.

#### 5.1.3. Cortisol Conclusions

In contrast to the associations found between neurophysiological signals and PA, our review revealed no consistent relationship between PA and cortisol levels. The absence of a direct relationship suggests that the calm and collected attributes of PA are unlikely to be related to a blunted HPA axis stress response. However, elevated SNS function has been linked to increased aggression ratings at lower cortisol reactivity [148] and to internalizing and externalizing problems at a higher basal cortisol level [149]. Although neither study specifically quantified PA, these findings highlight that asymmetries between SNS and HPA axis responses may be associated with atypical aggressive behavior in youths. Future studies may accordingly consider investigating cortisol levels in combination with other biomarkers to reveal potential multi-system connections between specific hormones and PA.

### 5.2. Testosterone

Testosterone is the primary male androgen and has been implicated in the masculinization of neural circuits during early development, as well as in the mediation of male and female social behaviors during adolescence [150,151]. The relationship between testosterone and aggression has been well established in animal models [152,153]; however, data from human studies are limited [154]. Males frequently endorse higher PA scores than females [34,41,79], and males may be prone to age-related PA score increases during the pubertal years [155]. Given these associations, it is possible that testosterone may play a role in the development of PA.

#### 5.2.1. Child and Adolescent Findings

Several studies have examined testosterone and PA in child and adolescent samples. One group measured aggression and salivary testosterone in youths aged 11 to 12 years [72]. Testosterone level did not predict PA or RA; however, as this study mainly sampled African American subjects of a narrow age range, these findings may not be generalizable to other ages or ethnicities. Among male adolescents, early work by Olweus and colleagues [58] reported a weak association between plasma testosterone and peer-rated unprovoked, aggressive behavior. Path analysis revealed no direct causal relationship with PA but instead identified testosterone as a positive predictor of RA [59]. Other studies have similarly found no association between salivary testosterone and PA in clinical populations of ASD and ODD/CD adolescent boys [73]. In contrast, a longitudinal study of North American males reported a positive significant association for both PA and RA and salivary testosterone at 16 years of age but not at ages 13 or 21 years [62]. The authors suggested that around the 16^th^ year, in the midst of puberty, the influence of testosterone on PA and RA is perhaps greatest and may be less relevant at different time points during development. Notably, PA and RA were highly correlated (*r* = 0.79) and not individually controlled for. As a result, any relationship between testosterone and aggression may not be specific for PA. Table 2 reflects the relationships that have been reported between testosterone and PA in child and adolescent studies.

#### 5.2.2. Adult Findings

Other authors have evaluated the link between testosterone and crimes driven by PA [60,61]. As reported in an investigation of 692 male prisoners, lower salivary testosterone predicted less violent and more covert criminal acts such as burglary, theft, and drug-related offenses [60]. While these results did not directly implicate PA, the authors suggested that higher testosterone may be related to a characteristic pattern of misbehavior that is indicative of PA. Among incarcerated men, higher salivary testosterone has also been positively correlated with the commission of premeditated murder, where the victim was known ahead of time [61]. Although it may be tempting to infer a positive association between PA and testosterone based on these findings, there is insufficient information on other potential offending patterns of these individuals that may be more driven by RA.

Unlike studies that infer PA based on past behavior, Carré and colleagues [65] compared real-time PA and salivary testosterone reactivity using the PSAP in adult males. Although increased testosterone was related to RA, no associations between testosterone and PA were present. These findings suggest that testosterone modulation related to in-the-moment aggression is more likely to be associated with RA than PA, which coincides with the studies reviewed above. Testosterone and PA findings in adults are reported in Table 2.

### 5.3. Other Hormones

One study to date has evaluated the relationship of PA and female sex-specific hormones (Table 2). Peters and coauthors [74] quantified PA and the ovulatory hormones estradiol and progesterone in a small sample (*n* = 15) of naturally cycling women with BPD. PA, measured using the RPQ, was highest during the follicular and ovulatory cycle phases, where progesterone was lowest. These findings were based on a small data set but may be pertinent for at-risk females with premenstrual dysphoric disorder, who are prone to increases in anger and irritability during normal cyclical hormonal changes [156]. Investigations into other female clinical populations that present with high aggression would also be beneficial.

## 6. Twin Studies

Behavioral genetic studies have documented that aggression has moderate to strong genetic underpinnings [157,158]. However, the genetic determinants of complex social behaviors such as PA are most likely due to multiple genetic influences. Phenotypic comparisons of monozygotic and dizygotic twin pairs can be used to estimate the heritability of a behavior of interest, namely PA, when the specific genes are unknown. In biology, heritability characterizes the resemblance of related individuals for a given characteristic [159]. Twin studies assume that monozygotic twin pairs share 100% of their genes and that dizygotic twins share 50%, thus making it possible to estimate the relative contribution of genetic factors, shared environmental factors (e.g., family environment), and non-shared environmental factors (e.g., peer groups that may differ between twins) [160]. Compared to genetic studies that identify and assess specific genes, the twin design estimates latent (e.g., unmeasured) factors.

To date, six twin studies that investigated PA as one aim have been conducted. These investigations are encompassed in two large longitudinal studies: the Quebec Newborn Twin Study [78,82,83] and the University of Southern California Risk Factors for Antisocial Behavior Twin Study [79,80,81] (Table 3). Data from each of the studies were fit to a variant of the Cholesky decomposition model that estimates the contributions of latent additive genetic (A), latent shared environmental (C), and latent non-shared environmental (E) factors.

Brendgen and colleagues [78] measured cross-sectional PA and RA in 6-year-old twins using the TRI. They reported that genetic effects accounted for 41% of the variance in PA; however, the majority of genetic effects (34%) were due to physical aggression, an underlying form of aggression common to PA and RA. Non-shared environmental effects unrelated to physical aggression accounted for 51% of the variation in PA and shared environmental effects exerted no significant influence. Only a fraction (6%) of the non-shared environmental effects could be jointly attributed to PA and RA, suggesting that aggression is influenced largely by non-shared environmental effects specific to PA or RA and only to a very small degree by genetic influences.

The Quebec Newborn Twin Study was also used to longitudinally assess PA throughout childhood (ages 6 to 12 years). Using a time-specific general latent factor model, Paquin and colleagues [157] reported that genetic factors explained approximately 50% of PA with moderate genetic stability from ages 6 to 12 years. However, 39–45% of genetic factors were common to PA and RA and less than 10% were unique to PA. In a subsequent study, a multivariate latent growth curve model was employed to evaluate the intra-individual development of PA over the same longitudinal time frame [83]. The authors determined that genetic factors common to PA and RA accounted for 64% of the variation in baseline PA (at age six years), while genetic influences that were unique to PA contributed to 43% of the longitudinal variation. Given that the genetic factors influencing baseline and developmental levels were different, the authors suggested a genetic maturation hypothesis. The hypothesis postulates that genetic factors associated with the inter-individual variation in developmental change are independent of baseline levels of PA and RA.

The California Risk Factors Study similarly evaluated latent PA and RA but instead used the RPQ and collected data across multiple informants (teacher, parent, and self-reports). An initial investigation compared cross-sectional PA and RA using the three report types in twins aged 9–10 years [79]. PA showed heritability rates of 32% and 45% according to parent and teacher reports, respectively, where male PA scores were higher than female scores across all informants. Self-report scores for PA indicated 50% heritability in males, yet 0% for females. The authors reasoned that biological variables with a strong genetic loading may be more pronounced for male self-report than for females. With the exception of female self-report scores, the heritability of PA was overall greater than that of RA. PA and RA were moderately to highly correlated within each report type (*r* = 0.46 to 0.80), but correlation scores between reporter types were low (*r* = 0.18 to 0.26), thus precluding the researchers from aggregating scores. Child self-report showed the greatest fit, according to confirmatory factor analysis, indicating that children themselves may be able to more precisely distinguish and self-report on PA compared with teachers or parents.

Longitudinal analysis of the California Risk Factors Study further determined that PA could be attributed primarily to genetic influences (63%), according to parent reports completed during late childhood (9 to 10 years old) and early adolescence (11 to 14 years old) [80]. Stability of PA across both sampling periods was largely explained by a common genetic factor that accounted for 85% of its stability, compared with RA, in which a common genetic factor only accounted for 48% stability. Analysis further determined that genetic influences for PA become increasingly more important over time. Notably, this research found differential stabilities between PA and RA, suggesting that PA is more stable over time and that RA is influenced to a greater degree by environmental factors.

Lastly, Bezdjian and colleagues [81] assessed the covariance of psychopathic traits and aggression in children aged 9 to 10 years old using child and parent reports. The researchers identified an association between psychopathic personality traits and PA, but findings were only significant for child reports. Supported by their colleagues [79], Bezdjian et al. [81] suggested that these results could reflect children having greater insight into their own behavior, at least in comparison to their parents’ perception. Child-reported data revealed that heritable influences accounted for 18% of PA, a modest finding when compared to the other twin studies reviewed. However, the study findings identified unique genetic and non-shared environmental influences for PA, RA, and psychopathic traits, suggesting independence among the behaviors and constructs. Ascertaining the covariance between complex behaviors, such as antisociality and aggression, may be beneficial in identifying susceptibility genes.

Importantly, the twin studies reviewed highlight that PA has a genetic basis that can be differentiated from RA. In fact, PA was reported as having greater heritability and being more genetically stable over the course of childhood and adolescence when compared with RA. PA heritability estimates ranged from 0% to 64%, reflecting both the relative influence of reporter type and the inherent heterogeneity of trait PA. PA and RA were highly correlated, and genetic influences common to both aggression types accounted for more than 50% of the variance, suggesting that the unique genetic contribution of PA is considerably less pronounced. Longitudinal findings show that heritable and environmental factors dynamically influence developmental PA, indicating that there is likely variation in susceptibility genes and their respective roles over the course of childhood, adolescence, and, presumably, into adulthood. Nonetheless, these findings did not take into account whether individuals had a psychiatric diagnosis and are, thus, limited in their applicability to clinical samples.

## 7. Molecular Genetic Studies

While latent analyses characterize the heritability of aggression, the relationship of specific gene variants to aggression has also been widely investigated. Most work has focused on the genetic underpinnings of reactive and impulsive forms of aggression. Hence, there is a need to identify genes associated with PA. Monoamine oxidase A (*MAOA*) is the most extensively researched gene related to aggression to date [161]. Located on the X chromosome, *MAOA* encodes for a key enzyme that degrades several monoamine neurotransmitters, such as serotonin, dopamine, and norepinephrine [162,163]. The upstream variable number of tandem repeats (uVNTR) of *MAOA* is a 30 bp region shown to have 2, 3, 3.5, 4, or 5 repeat motifs [164]. The 2R and 3R variants are associated with low transcriptional activity (*MAOA-L*), whereas 3.5R and 4R variants are associated with high transcriptional activity (*MAOA-H*) [165,166]. *MAOA-L* alone, or in combination with childhood maltreatment, has been associated with antisocial and aggressive behavior [167,168]; nonetheless, selected investigations have also reported a relationship between increased aggression and the *MAOA-H* genotype in males [169,170].

One group evaluated *MAOA* uVNTR and PA (measured by the aggressiveness subscale of the Freiburg Personality Inventory) [94] in a community sample but reported no direct relationship with *MAOA* genotype [84]. While an association between *MAOA-L* and RA for both males and females was detected, the authors contended that the low-activity genotype contributed to increased aggressive reactivity to provocation and not to aggressive behavior as a general construct. Among males with antisocial personality disorder (ASPD), positive associations between PA and *MAOA-H* [85], as well as for PA and *MAOA-L*, have been reported [13]. The former study suggested that *MAOA-H* may interact with environmental factors (e.g., childhood physical abuse) to increase the risk of offending and predatory violence, whereas the latter investigation determined that brain resting state conditions were also related to genotype. It should be noted, however, that these studies employed relatively small sample sizes. The latter study also reported that the *MAOA-H* genotype may affect brain functioning in ASPD individuals [13], a finding that is further discussed in the neuroimaging section of this review. In addition to the uVNTR *MAOA* genetic variants, one group evaluated the relationship between the *MAOA* T941G genetic polymorphism and PA in a relatively large sample (*n* = 1399) of Chinese Han adolescents but reported no significant association [87]. Table 4 reflects the genetics of *MAOA* and all other molecular genetic findings that included an index of PA.

Other research groups have investigated the catechol-O-methyltransferase (*COMT*) gene, whose gene product is known to catabolize dopamine, especially in the prefrontal cortex (PFC) [171,172]. Among *COMT* polymorphisms, Val158Met has been identified as a low-activity polymorphism capable of reducing dopamine degradation by up to four-fold [173]. Investigations of male forensic inpatients [88] and community samples [87] have found no significant associations between PA and this genetic variant.

Dopamine has been implicated in processes that are relevant to PA, such as reward appraisal and goal-setting [127,174]. The D4 dopamine receptor (*DRD4*) gene is widely expressed throughout the brain and has been linked to violence and aggression [175,176]. The VNTR polymorphism in exon 3 ranges from 2 to 11 tandem repeats, whereas the 7R genotype has shown less dopamine binding compared with 2R and 4R genotypes [177]. Increased prevalence of 5R/5R, 5R/7R, and 7R/7R *DRD4* genotypes have been reported among proactively violent male offenders, as determined by conviction record and serial interviews [86]. The *DRD4* 7R genotype has also been linked to novelty-seeking, risk taking, and under-reactivity to unconditioned aversive stimuli, as well as efficient problem solving among healthy males [178], perhaps providing evidence that PA may be more prevalent in males with this genotype.

Several studies have explored the relationship between PA and social behavior genes. The oxytocin receptor (*OXTR*) gene is expressed in the hypothalamic regions of the brain among other brain regions and other peripheral tissues that encode for the binding site of the neuropeptide oxytocin. Oxytocin is involved in lactation and parturition, as well as prosocial behavior, social affiliation, and pair bonding [179]. One study of community adolescents investigated longitudinal PA (measured using the SRASBM between the ages of 13 to 18 years) in relation to five *OXTR* single-nucleotide polymorphisms (SNPs) (rs1042778, rs2254298, rs53576, rs4686302, and rs237915) that have been linked to antisocial behavior [91]. No significant associations were identified between PA and the pre-selected SNPs; however, there are other common *OXTR* SNPs that this study did not measure. Other investigators have examined associations between aggression and genes of the endogenous *μ*-opioid system, a neurochemical system that has been implicated in the regulation of social rewards and affiliation, as well as pain perception [180,181]. Among healthy adult males, the functional A118G polymorphism of the *μ*1-subtype opioid receptor (*OPRM1*) gene showed no direct association with PA [92]. The study tested 72 subjects and may not have been adequately powered to detect associations.

To date, only one genome-wide association study (GWAS) has analyzed the genetic underpinnings of PA and RA. GWASs detect associations between variants at genomic loci (e.g., SNPs) and complex traits, such as aggression, by scanning the genomes of many subjects [182]. Van Donkelaar and colleagues [89] analyzed data from a large sample of healthy children and adolescents (*n* = 18,988) and identified two candidate genes that were associated with risk of aggression, but no associations with PA were identified. In an attempt to better understand the genetic basis of PA, a gene-set association analysis aggregated common genetic variants in or within 100 kb flanking regions of serotonergic, dopaminergic, and neuroendocrine signaling genes [93]. Using their community-based, mixed-sex, adult sample, the investigators noted that PA was greater in males, but no genetic variants were specifically implicated. The researchers opined that analyses taking into account sex differences are necessary for parsing the genetic basis of PA.

Yang and colleagues [90] investigated the frequencies of selected short tandem repeat (STR) loci on the Y chromosome (Y-STR) among 271 males who were imprisoned for PA-related behavior (assault *n* = 54; robbery *n* = 204; murder *n* = 13) with no history of psychiatric illness or substance use disorders. Y-STR are routinely utilized for male-specific forensic DNA identification, as these mutations are transmitted without recombination from father to son [183,184]. Although the underlying function of Y-STRs remains presently unknown, sex determination and the subsequent production of testosterone are localized to the Y chromosome [185] and, as noted previously, testosterone level has shown some relationship with PA [62]. The commission of premeditated crimes was shown to occur at a higher rate in carriers of STR loci DYS533 (14 repeats) or DYS437 (14 repeats), suggesting a potential genetic link to PA in these males. Unique associations between DYS448 (18 and 22 repeats) and DYS456 (17 repeats) have also been identified for impulsive–aggressive criminal offenders [186], perhaps suggesting that the relationship between these polymorphisms and aggression is more general in nature. It is notable that subjects from both STR studies were entirely Chinese Han males, and it is unknown if these findings are generalizable to other ethnicities.

Much of the genetic literature on PA identified in this review comes from genetic association studies, which do not always yield reproducible effects in part due to small sample sizes. Evidence that *MAOA* and *DRD4* genetic polymorphisms may be related to PA could pertain to their control of neurotransmitters implicated in aggressive behaviors [167,187]. However, these relationships are far from clear, and GWASs are necessary to shed further light on the relationship between gene effects and PA. Epigenetics may also be a fruitful field, although to the best of our knowledge, there has never been a study that has investigated epigenetic mechanisms in relation to PA.

## 8. Neurophysiology

Event-related potentials (ERPs) are rapidly changing electrophysiological states that reflect neuronal functioning and the associated brain processes [188]. Derived from electrical potential gradients measured at the scalp, ERPs evaluate changes in the state of electrical activity in response to an experimental stimulus [189]. The most frequently used measures are the amplitude of a wave and the time to the peak (latency), where peaks at different times correspond to unique indicators of neurological function.

The P3 wave indexes attentional resource allocation and is characterized by a positive peak 300 to 700 ms after stimulus presentation. This wave has been frequently used in the study of antisocial traits and is believed to reflect interactions between the frontal cortex and hippocampal temporal/parietal junction [190,191]. P3 amplitudes among PA offenders have shown less interference in response to sad cues [97] and marginally longer latency (time to peak) when presented as an auditory stimulus [96], relative to a non-offender comparator group. Furthermore, impulsive and premeditated aggressors alike have shown similar P3 amplitudes across physical threat, social threat, and neutral words, perhaps indicating less efficient processing and a tendency to recognize aversive stimuli as neutral [98]. Others have reported no relationship between PA and P3 amplitudes among impulsive and non-impulsive aggressive offenders [95].

The N2 is a negative frontocentral wave that peaks 200 to 350 ms after stimulus onset in tasks demanding attention or response inhibition [192]. Using mediation analyses, one group reported an association between PA and reduced N2 during the decision phase of the TAP for impulsively violent adolescent offenders [99]. A reduction in general inhibitory control, as indexed by the N2, is likely more indicative of the impulsive, reactively aggressive traits of the study participants and less so to PA specifically. Despite limited neurophysiological studies on ERPs, further study of perturbations of P3 and N2 waves in relation to the commission of PA may be warranted. Table 5 summarizes the relationship between PA and neurophysiological findings.

## 9. Neuroimaging

Research on the neural correlates of aggressive behavior has historically implicated the frontal and temporal lobes, as these regions control the way we perceive, process, and react to our surrounding environment [193]. While this review has repeatedly emphasized that PA and RA are often correlated [194], different motivations for proactive versus reactive behaviors underscore potentially unique neural underpinnings, hence the need to evaluate each aggressive subtype individually [195]. As previously mentioned, RA requires a provoking stimuli, and those high in RA are quick to anger. Reactively aggressive behavior has been associated with dysfunction in the orbitofrontal cortex and hyperactivity of the amygdala, regions that contribute to emotional processing and threat assessment, respectively [196,197]. On the other hand, PA is goal oriented, where individuals high in PA are more likely to anticipate positive outcomes as the result of their aggression [198,199]. These clinical observations suggest that the neural basis of PA may be associated with regions, such as the dorsolateral prefrontal cortex (DLPFC), which relate to planning and goal setting [197].

Over the last two decades, advances in brain imaging technologies have afforded opportunities to quantify brain volumes and investigate neural activity, in real time, related to emotional or cognitive processing. MRI is the most popular neuroimaging modality currently in use; however, positron emission tomography (PET), which uses ionizing radiation to capture imaging data, has also been utilized. Structural MRI (sMRI) can measure region-specific brain volumes using a three-dimensional grid of voxels, while functional MRI (fMRI) utilizes blood oxygenation level-dependent (BOLD) contrast to localize neural activity to one or more regions [200], which may be measured at rest (resting-state fMRI (rsfMRI)) or during an experimental task. In addition, proton magnetic resonance spectroscopy (^1^H-MRS) is an imaging modality used to quantify selected metabolite concentrations.

### 9.1. PET

Early work by Raine and colleagues [102] compared brain function in groups of affective murderers, predatory murderers, and healthy subjects using fluorodeoxyglucose (FDG) PET. Predatory murderers showed increased glucose metabolism in right subcortical regions but were otherwise similar to the healthy subjects. Affective murderers likewise showed increased right subcortical function but also demonstrated reduced lateral prefrontal functioning and reduced right prefrontal to subcortical ratios, relative to the healthy subjects. The researchers postulated that increased right subcortical functioning predisposes one to aggressive behaviors and that the regulation of aggressive impulses among predatory subjects, as inferred by the prefrontal to subcortical function ratio, may reflect relatively improved emotional control compared to affective murderer subjects. Table 6 summarizes PET findings related to PA.

### 9.2. Structural MRI

Some research [103,106,112], but not all [108], has revealed negative associations between PA and right amygdala volume determined by voxel-based morphometry. One longitudinal study of males with varying histories of violence assessed PA from childhood to adulthood [106]. According to sMRI images obtained at 26 years of age, right amygdala volume was negatively associated with adolescent PA, while both the left and right amygdalae volumes were negatively correlated with a history of premeditated acts in adulthood. These findings suggest an association between smaller amygdala volume and a longstanding developmental history of heightened PA, dating back to early childhood. Nonetheless, right amygdala volume has also been negatively associated with impulsivity [106] and hostility [103], perhaps indicating that the functionality of this structure may also relate to RA.

Another group evaluated the neural correlates of PA and RA among cohorts of typically developing and ODD/CD youth using several regions of interest [112]. Across the entire sample, negative associations were reported for amygdala volumes and PA, as well as right insular volume and RA. These findings persisted after controlling for confounding variables, such as age, sex, and total brain volume. The amygdala, as previously mentioned, is implicated in threat assessment, while the insula is involved in human awareness [201]; both regions belong to the frontolimbic network. Thus, it is possible that connectivity among these regions may account for some of the observed overlap between PA and RA [202]. Furthermore, the involvement of limbic structures in both PA and RA highlights the potential importance of emotional processing, irrespective of aggressive subtype [203].

Involved in emotional and behavioral regulation, the anterior cingulate cortex (ACC) is a subcortical limbic structure that may play a role in PA [204,205]. PA has been negatively associated with left ACC thickness among male and female adolescents [107] and positively correlated with left ACC volume in adult males [108]. Aggression as a general construct has also been linked to reduced ACC thickness and volume in children and adolescents [206,207], whereas increased volume has shown a relationship with improved behavioral regulation in adults [208]. The cavum septum pellucidum (CSP), which is a slit-like space in the septum pellucidum that forms during gestation and typically closes by six months postnatal, is a relatively common neuroanatomical variant that is thought to reflect abnormal development of limbic structures [209]. Comparisons of CSP size (large CSP (>4 mm) versus small CSP) and PA among DBD and non-DBD male youths found positive associations between large CSP, DBD diagnosis, and PA [104]. PA was unrelated to CSP size within the DBD sample, suggesting that CSP size differences may relate more to the DBD phenotype.

Correlational analyses have linked PA to increased grey matter density (GMD) in the bilateral DLPFC and reduced GMD in the posterior cingulate cortex (PCC) in college students [111]. The DLPFC is implicated in working memory and organization related to goal setting, whereas the PCC is associated with internally directed cognition and is integral to operation of the default mode network (DMN) [197,210,211]. The DMN is most active when the brain is in a state of wakeful rest in the absence of any attention-demanding tasks; structurally, it consists of the medial prefrontal cortex (MPFC), PCC/precuneus, and angular gyrus [212]. In support of these findings, transcranial direct current stimulation of the left DLPFC has been shown to reduce PA in healthy men [14], while deactivation of the DMN has shown an association with CU traits and aggression in adolescents [213,214].

Lower brain volumes in the right insular, left planum polare, and bilateral precuneal regions have been observed among males with schizophrenia with a history of premeditated violent acts [12]. These brain structures are implicated in awareness, auditory and visual processing and one’s ability to understand the mental states of others and respond appropriately, also known as Theory of Mind (ToM) [197,215,216,217]. Notably, ToM may positively predict PA, as the ability to recognize the mental states of others can be advantageous when aggressing others for personal gain [218,219]. The findings of Kuroki and colleagues [12] are, nevertheless, limited in that the premeditated acts by which PA was measured occurred during acute psychosis in many subjects. Thus, results may be more applicable to the effects of psychotic symptomatology rather than PA. Figure 2 and Table 6 summarize the sMRI findings of PA.

### 9.3. Functional MRI Studies

Right amygdala responses to fearful expressions, as assessed by fMRI, have been shown to negatively predict PA in youths with high CU traits [105]. Interestingly, it was suggested that amygdala hypo-activity may represent an intermediate phenotype that mediates the association between CU traits and PA, where these traits stem from deficient amygdala responses to distress. Using a seed-to-voxel approach, another study of brain function at rest pointed to increased left amygdala–precuneus functional connectivity (FC) for PA in ODD/CD and healthy adolescents [113]. Although group differences were not detected, these findings may help explain how amygdala perturbations possibly influence DMN function related to PA. PA has been associated with decreased FC between the DLPFC and the bilateral inferior parietal lobes (IPL) and PCC, as well as FC between the PCC and MPFC/ACC, bilateral IPL, and precuneus among university students [111]. The results of this study suggested that multiple networks may be related to PA among non-clinical participants with low PA scores.

Among ASPD males with a history of violent offending, FC related to PA was shown to vary in a *MAOA* genotype-specific manner for high- (*MAOA-H*) and low-activity (*MAOA-L*) genotypes [13]. Increased PA in violent *MAOA-L* males with ASPD was strongly correlated with FC between the precuneus and angular gyrus. Both regions are nodes of the DMN that have been implicated in the genesis of externalizing behaviors [220,221]. Another study of violent offender- and non-offender males reported a positive association between PA and FC in the MPFC and a negative association between PA and superior temporal sulcus (STS) BOLD activity across both groups [111]. The STS has been implicated in rumination [222] and ToM [223] which, as previously discussed, may show some relationship with PA. fMRI findings related to PA are summarized in Figure 2 and Table 6.

### 9.4. ^1^H-MRS Studies

Very little is known about the localization, concentration, and function of brain metabolites as they relate to PA. The primary mammalian excitatory neurotransmitter glutamate has been more recently linked to aggression with respect to aberrant signaling within the fronto-amygdala-striatal region [224,225]. One multi-site imaging study of 233 participants evaluated the relationship between PA and regional glutamate concentration using ^1^H-MRS in DBD youths [109]. An inverse association was identified between PA and glutamate levels in the left dorsal striatum, an area thought to contribute directly to decision making [226]. Interestingly, CU traits were positively associated with glutamate in the ACC, suggesting dissociable correlates of PA and CU in DBD youths. Increased aggression has also been associated with elevated DLPFC glutamate metabolites in ASPD males [227], indicating perhaps that prefrontal glutamate levels may be relevant to PA, although the latter study did not differentiate between PA and RA. Data from the singular ^1^H-MRS study on PA are reflected in Figure 2 and Table 6.

### 9.5. Imaging Conclusions

Much of the imaging literature on PA implicates the frontal and limbic regions, nodes of the DMN, and the amygdala. Morphometric findings suggest that reduced amygdala volumes and perturbations to the ACC may be linked to PA, while fMRI data point to amygdala hypo-activity and alterations of the DMN. At present, the specific associations between PA and the functioning of the DMN are unclear as both negative [111] and positive [13] findings were identified in our review. It is possible that these conflicting results are due to the variability in PA between community and aggressive cohorts, or that the PA–DMN relationship is specific to *MAOA-L* genotypes [13]. The DMN is also notable as it has been empirically linked to ToM [228] and thus may hold clues about how having insight into the vulnerability of others can be advantageous when proactively aggressing for personal gain. Overall, the hierarchical organization of the brain makes it apparent that multiple brain systems are likely associated with PA, which may vary according to study population sampled or the severity of PA.

## 10. Conclusions

This review was undertaken to assimilate and evaluate the biological correlates of PA in the extant literature. As discussed, PA is typically enacted with an end goal that is beyond the immediate purview of the aggressive act itself and is thus premeditated in nature. We surmised that the literature would support findings of the cool, collected, and unemotional attributes of PA—in contrast to the emotional, impulsive, and hyper-aroused states that are characteristic of RA.

Perhaps the strongest findings were the relationship between PA and SNS under-arousal. For example, PA was strongly associated with dampened RHR, and moderately so with lower HRR and SCR. Reduced SCR was notably related to deficient fear conditioning, indicating that PA may be associated with blunted assessment of aversive stimuli that can inform future behavior [36,42]. Nonetheless, associations directly relating PA to PNS function were largely absent. Likewise, there were few data to support an association between PA and cortisol levels, suggesting that dampened HPA axis mediated stress responses may not comprise a neuroendocrinological correlate of PA.

The twin literature indicated a moderately strong genetic basis of PA, with unique influences for both baseline PA and longitudinal development of PA. Genetic association studies, in contrast, provided limited support for a relationship between specific gene variants and PA. Genetic polymorphisms of *MAOA* and *DRD4* genes showed some connection with PA, suggesting that dopaminergic neurotransmission may be relevant to enactment of PA (Table 4). Compared with other psychiatric disorders, such as schizophrenia or major depressive disorder, where the neurochemistry has been extensively investigated [229,230,231,232,233,234], very little is known about the neurochemical correlates of PA. Given the lack of clinical data demonstrating the efficacy of psychotropic medications on PA [235], the elucidation of novel signaling pathways underlying PA is the logical next step to developing potential rational pharmacological interventions.

Imaging studies have implicated structural and functional changes of the amygdala in PA. These findings are unsurprising, as a relatively large body of evidence points to the involvement of the amygdala in aggression as a general construct [236]. Structurally, volumetric reductions in the amygdala were linked to higher PA. Functionally, amygdala hypo-activity may represent intermediate phenotypes that mediate the relationship between antisocial behavior and trait PA [105]. Several studies implicated regions of the DMN. Both positive [13] and negative [111] associations between PA and DMN connectivity were identified in the literature, indicating a need for more investigations to clarify the relationships between PA and individual nodes, as well as overall DMN activation/deactivation.

One of the greatest challenges of characterizing the neurobiology of PA is recognizing that PA is a multi-faceted concept, encompassing attributes such as goal orientation, lack of provocation, and low arousal. Although multiple instruments have been developed to quantify PA, to the best of our knowledge, no evaluation tool to date is able to comprehensively assess PA across its substituent components. For example, the RPQ characterizes PA primarily by violent behaviors (or the threat thereof) to establish dominance, whereas, instruments such as the SRASBM have been validated to quantify relational PA, which is more covert in nature. As demonstrated in our review, relational PA does not appear to conform to the same neurobiological correlates as overt forms of PA (e.g., [37,43]). With respect to the heterogeneous nature of PA, it is reasonable to assume that brain regions implicated in aggression in the absence of provocation might differ from premeditated acts. An important aspect of PA research, therefore, must take into account the inherent complexity and diverse motivations underlying PA.

Several limitations of this review should be addressed. Notably, our review did not include animal studies. Although PA can be measured in rodent models [237], animal behavioral paradigms have been criticized for lacking validity, in particular for CU traits [238]. We also excluded studies that quantified the PA subtype known as appetitive aggression (AA). Described as blood lust or hedonistic aggression, AA is known to occur among combatants, gang members, and individuals with chronic or egregious exposure to violence [239]. AA is driven by pleasure of the aggressive act in its own right and is typically associated with excitement and high arousal. Thus, we deliberately excluded these studies to limit heterogeneity in an already diverse area of research. Lastly, studies that measured PA in reference to experimental treatments, such as transcranial direct current stimulation [14] or pharmacological medications [240], were not included. The aims of this review were to identify and interpret the biological correlates of PA and not to evaluate treatments and therapies intended to manage PA.

In conclusion, we report that the neurobiological correlates of PA represent a diverse area of study that goes far beyond the “cold blooded” characterization. The majority of PA findings published to date are based on community samples; thus, bridging the gap between “normal” and pathological PA remains an important research priority. As our review revealed that multiple PA endophenotypes likely exist, future research should focus on defining the corresponding genetic and neurological predictors in an attempt to further disentangle PA as a complex behavior. Furthermore, elucidation of the specific neural structures, networks, and implicated neurotransmitters in PA could only enhance the translation of findings to treat PA with more precision, utilizing behavioral and pharmacological methods.

## Figures and Tables

**Figure 1 brainsci-11-01412-f001:**
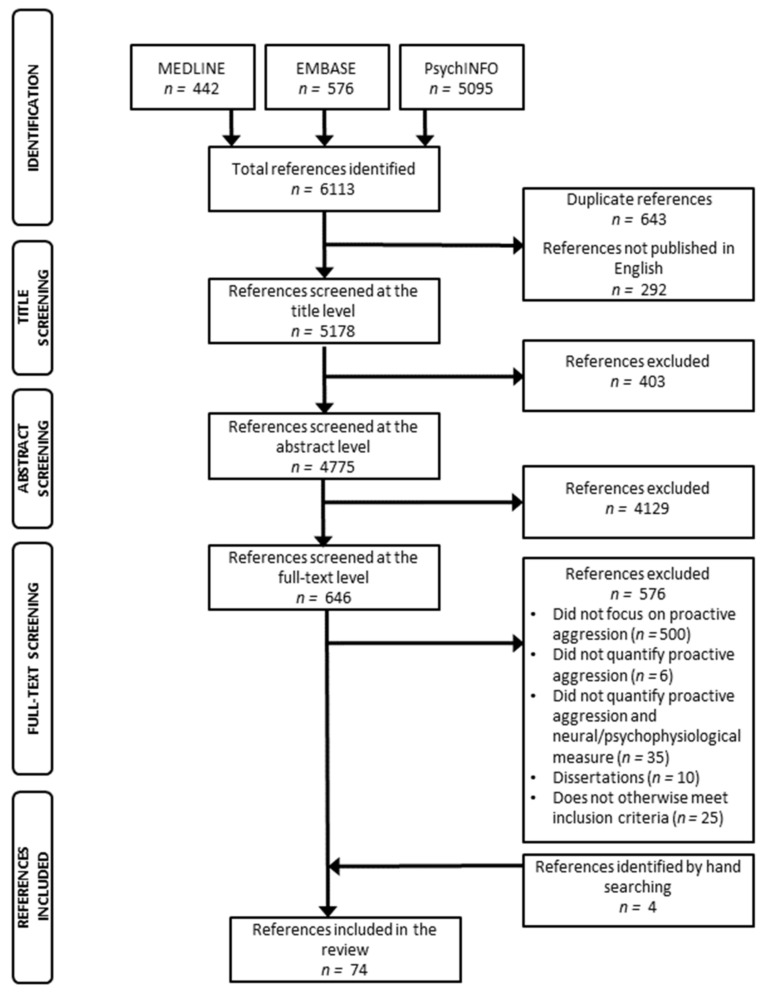
Article screening flow diagram.

**Figure 2 brainsci-11-01412-f002:**
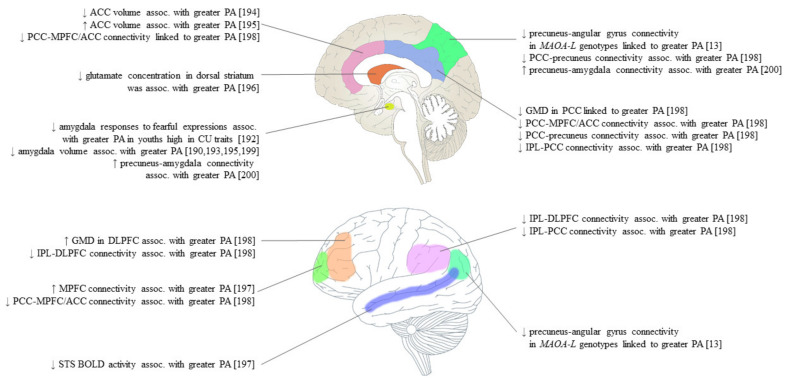
Neuroimaging findings of proactive aggression (PA), as quantified by the Reactive–Proactive Questionnaire [11], grouped by brain region. ACC, anterior cingulate cortex; Assoc., associated; PCC, posterior cingulate cortex; MPFC, medial prefrontal cortex; CU, callous–unemotional; *MAOA-L,* monoamine oxidase A low-activity genotype; GMD, grey matter density; IPL, inferior parietal lobe; DLPFC, dorsolateral prefrontal cortex; STS, superior temporal sulcus; BOLD, blood oxygenation level-dependent.

**Table 1 brainsci-11-01412-t001:** Physiological findings according to resting heart rate (RHR), heart rate reactivity (HRR), heart rate variability (HRV), resting skin conductance (RSC), skin conductance reactivity (SCR), resting respiratory sinus arrhythmia (RRSA), and respiratory sinus arrhythmia reactivity (RSAR) of included studies.

Study	Year	*n*	Age	Sex	Region	PA Instrument	RHR	HRR	HRV	RSC	SCR	RRSA	RSAR
Crozier et al. [34]	2008	585	17	M/F	North America	RPQ	***r* = −0.15 (M)** *r* = 0.11 (F)	*r* = 0.14 (M) *r* = 0.09 (F)					
Hubbard et al. [15]	2010	36	10	M/F	North America	Paradigm task		***r* = −0.35**			***r* = −0.58**		
Scarpa et al. [35]	2010	42	10	M/F	North America	R-PRPA	*β* = 0.02 *d* = 0.03		***β* = 0.43** ***d* = 0.65**	***β* = 0.46** ***d* = 0.69**			
Bobadilla et al. [36]	2012	122	19	M/F	North America	TAP					***β* = −0.23**		
Murray-Close and Rellini [37]	2012	83	22	F	North America	SRASBM		*r* = −0.14					*r* = 0.13
Muñoz Centifanti et al. [38]	2013	85	16	M	North America	CRTT		*r* = −0.06			*r* = −0.13		*r* = 0.12
Portnoy et al. [39]	2014	335	16	M	North America	RPQ	***r* = −0.13**	***r* = −0.11**					
Raine et al. [40]	2014	334	13	M/F	Asia	RPQ	***r* = −0.18**						
Xu et al. [41]	2014	183	8	M/F	Asia	TRI	***r* = −0.27**					*r* = 0.03	
Gao et al. [42]	2015	329	17	M/F	North America	RPQ					***d* = −0.47**		
Wagner and Abaied [43]	2015	168	19	M/F	North America	SRASBM					*r* = −0.08		*r* = −0.12
Zhang and Gao [44]	2015	84	22	M/F	North America	RPQ						*β* = 0.12	*β* = −0.08
Schoorl et al. [45]	2016	102	10	M	Europe	IRPA	***β =* −0.38**	***β =* −0.48**	***β =* 0.41**	*β* = −0.10	***β* = −0.34**		
Wagner and Abaied [46]	2016	180	20	M/F	North America	SRASBM					*b* = −0.02		
Murray-Close et al. [47]	2017	247	19	M/F	North America	SRASBM					*b* = 0.02		*b* = 2.02
Kassing et al. [48]	2018	188	11	M/F	North America	TRI				*β* = 0.00		*β* = 0.00	
Moore et al. [49]	2018	35	11	M/F	North America	Paradigm task				*β* = −0.46	***β* = -1.26**	*β* = −0.02	***β* = 0.11**
Ungvary et al. [50]	2018	58	14	M/F	North America	36-itemself-report ^§^						*β* = 0.04	*β* = −0.13
Armstrong et al. [51]	2019	509	20	M/F	North America	RPQ	*r* = −0.03 (M) ***r* = 0.13 (F)**	*r* = 0.00 (M) *r* = −0.11 (F)		*r* = −0.06 (M) ***r* = 0.15 (F)**	***r* = −0.18 (M)** *r* = 0.09 (F)		
Thomson and Beauchaine [52]	2019	104	20	M/F	Europe	RPQ						*b* = 0.16	
Hagan et al. [53]	2020	178	22	F	North America	RPQ							*β* = 0.10
Puhalla et al. [54]	2020	81	22	M/F	North America	TAP	***r* = −0.23**		*r* = 0.09				

Statistical findings shown in bold are significant according to *p*-values < 0.05; ^§^ [55]; *r*, Pearson correlation coefficient; *b*, unstandardized regression coefficient; *β*, standardized regression coefficient; *d*, Cohen’s d measure of effect size; RPQ, Reactive–Proactive Aggression Questionnaire [11]; R-PRPA, Revised Parent Rating Scale for Reactive and Proactive Aggression [56]; TAP, Taylor Aggression Paradigm [26]; SRASBM, Self-Report of Aggression and Social Behavior Measure [20]; CRTT, Competitive Reaction Time Task [57]; TRI, Teacher Report Instrument [4]; IRPA, Instrument for Reactive and Proactive Aggression [24].

**Table 2 brainsci-11-01412-t002:** Studies on hormones related to PA arranged according to year of publication.

Study	Year	*n*	Age	Sex	Region	PA Instrument	Hormone	Sample	Correlation/ Regression Coefficient	Findings
Olweus et al. [58]	1980	58	16	M	Europe	Peer-rated PA	Testosterone	Plasma	*r* = 0.16	PA was weakly correlated with plasma testosterone.
Olweus et al. [59]	1988	58	16	M	Europe	Peer-rated PA	Testosterone	Plasma	*r* = 0.21	PA was weakly correlated with plasma testosterone but no direct relationship was identified by path analysis.
Dabbs et al. [60]	1995	692	20	M	North America	Offending record	Testosterone	Saliva	*na*	More covert crimes (e.g., theft or drug offences) were associated with lower testosterone.
Dabbs et al. [61]	2001	230	20	M	North America	Offending record	Testosterone	Saliva	***r* = 0.35**	Higher testosterone was associated with premediated crimes, where the victim was known ahead of time.
van Bokhoven et al. [62]	2005	96	13, 16, 21	M	North America	TRI	Testosterone	Saliva	*na*	Analysis of variance determined testosterone was greater in high-PA boys (versus low-PA boys) at 16 years of age, but not at 13 or 21 years.
Kempes et al. [63]	2006	78	10	M/F	Europe	Dyadic play sessions and parent-report	Cortisol	Saliva	*na*	No significant association with PA.
Lopez-Duran et al. [64]	2009	73	7	M/F	North America	TRI	Cortisol	Saliva	*β* = −1.16	No significant association with PA.
Carré et al. [65]	2010	151	20	M	North America	PSAP	Testosterone	Saliva	*r* = 0.22	No significant association with PA.
Poustka et al. [66]	2010	245	15	M/F	Europe	VIRA-R	Cortisol	Plasma	***r* = −0.23 (M)** *r* = −0.06 (F)	Cortisol was negatively associated with PA in males.
Catherine et al. [67]	2012	89	10	M/F	North America	TRI	Cortisol	Saliva	***r* = −0.38**	Peer-nominated PA was negatively associated with afternoon cortisol.
Dietrich et al. [68]	2013	1961	11	M/F	Europe	YSR and ASBQ	Cortisol	Saliva	*β* = 0.10	No significant association with PA.
Johnson et al. [69]	2014	57	19	M/F	North America	SRASBM	Cortisol	Saliva	*r* = 0.05	No significant association with PA.
Stoppelbein et al. [70]	2014	158	10	F	North America	TRI	Cortisol	Plasma	***r* = −0.17** *β* = −0.11	Cortisol was negatively correlated with PA but path analysis determined no significant association.
Oberle et al. [71]	2017	151	11	M/F	North America	TRI and peer nominations	Cortisol	Saliva	***β* = −0.12**	Afternoon cortisol negatively predicted PA.
Chen et al. [72]	2018	445	12	M/F	North America	RPQ	Testosterone	Saliva	*β* = 0.05 (M) *β* = 0.05 (F)	No significant association with PA.
Ungvary et al. [50]	2018	58	14	M/F	North America	36-item self-report ^§^	Cortisol	Saliva	***β =* 0.24**	Cortisol was positively associated with PA when subjects were anticipating peer rejection.
Bakker-Huvenaars et al. [73]	2020	114	15	M	Europe	RPQ	Cortisol Testosterone Oxytocin	Saliva	*r* = 0.02 *r* = 0.16 *r* = 0.02	No significant association with PA.
Peters et al. [74]	2020	15	32	F	North America	RPQ	Progesterone	Saliva	*na*	PA was highest in the follicular and ovulatory cycle phases when progesterone was lowest.

Statistical findings shown in bold are significant according to *p*-values < 0.05; ^§^ [55]; *r*, Pearson correlation coefficient; *β*, standardized regression coefficient; *na*, not applicable; TRI, Teacher Report Instrument [4]; PSAP, Point Subtraction Aggression Paradigm [27]; VIRA-R, Vragenlijst Instrumentele En Reactieve Agressie [75]; YSR, Youth Self-Report [76]; ASBQ, Antisocial Behavior Questionnaire [77]; SRASBM, Self-Report of Aggression and Social Behavior Measure [20]; RPQ, Reactive–Proactive Aggression Questionnaire [11].

**Table 3 brainsci-11-01412-t003:** Twin studies that evaluate proactive aggression arranged according to year of publication.

Study	Year	*n*	Age(s)	Sex	Region	PA Instrument	PA Genetic Contribution	Findings
Brendgen et al. [78]	2006	344	6	M/F	North America	TRI	41%	The majority of genetic effects (34%) were due to physical aggression, which was common to PA and RA; genetic influences specific to PA were limited.
Baker et al. [79]	2008	1219	10	M/F	North America	RPQ	0% to 50%	PA exerted a greater genetic influence than RA, and child-report PA data showed the greatest fit among report types. Male PA scores were higher than female scores across all report types.
Tuvblad et al. [80]	2009	1241	10, 12	M/F	North America	RPQ	32% to 48%	PA becomes increasingly stable over time, compared to RA, which appears to be influenced more strongly by environmental factors.
Bezdjian et al. [81]	2011	1219	10	M/F	North America	RPQ	18% to 37%	PA was associated with psychopathic traits, but only for child-reported measures. Both heritable and non-shared environmental influences were found for PA and psychopathic traits, suggesting etiological differences in young twins.
Paquin et al. [82]	2014	1110	6, 7, 9, 10, 12	M/F	North America	TRI	39% to 45%	The contributions of unique PA influences were limited (0.2% to 9.4%), but factors common to PA and RA showed persistent associations during childhood.
Paquin et al. [83]	2017	1110	6, 7, 9, 10, 12	M/F	North America	TRI	47% to 64%	Genetic factors that influence baseline and developmental PA are independent of each other.

TRI, Teacher Report Instrument [4]; RPQ, Reactive–Proactive Aggression Questionnaire [11].

**Table 4 brainsci-11-01412-t004:** Proactive aggression molecular genetic findings according to year of publication.

Study	Year	*n*	Age	Sex	Region	PA Instrument	Gene(s)	Correlation/ Regression Coefficient	Findings
Kuepper et al. [84]	2013	239	23	M/F	Europe	FPI and Modified TAP	*MAOA*	*na*	No significant association between PA and uVNTR.
Kolla et al. [85]	2014	31	38	M	North America	RPQ	*MAOA*	***β* = 4.4**	PA was positively associated with *MAOA-H.*
Cherepkova et al. [86]	2015	586	39	M	Eurasia	Offending record	*DRD4*	***τ_(16)_* = 1.00**	PA was associated with 5R/7R and 7R/7R *DRD4* genotypes.
Zhang et al. [87]	2016	1399	12	M/F	Asia	RPQ	*MAOA* *COMT*	|*β*|s ≤ 0.07	No significant association with PA.
Kolla et al. [13]	2018	40	35	M	North America	RPQ	*MAOA*	*na*	PA was positively associated with *MAOA-L.*
van Dongen et al. [88]	2018	71	38	M	Europe	RPQ	*COMT*	*r* = 0.35	No significant association with PA.
van Donkelaar et al. [89]	2018	501	25	M/F	Europe	RPQ	GWAS	*r* ≤ 0.74	No significant associations with PA.
Yang et al. [90]	2018	763	32	M	Asia	Offending record	Y chromosome STR loci	*na*	PA was positively associated with STR loci DYS533 (14 repeats) and DYS437 (14 repeats).
Fragkaki et al. [91]	2019	323	13	M/F	Europe	SRASBM	*OXTR*	*b* = 221.4	PA was not significantly associated with the *OXTR* A118G polymorphism.
Weidler et al. [92]	2019	59	25	M	Europe	RPQ	*OPRM1*	*na*	No significant association with PA.
van Donkelaar et al. [93]	2020	395	24	M/F	Europe	RPQ	Gene-set association	*na*	No significant association with PA.

Statistical findings shown in bold are significant according to *p*-values < 0.05; *r*, Pearson correlation coefficient; *b*, unstandardized regression coefficient; *β*, standardized regression coefficient; *τ*, Kendall rank correlation coefficient; *na*, not applicable; FPI, Freiburg Personality Inventory [94]; TAP, Taylor Aggression Paradigm [26]; RPQ, Reactive–Proactive Aggression Questionnaire [11]; GWAS, Genome-wide association study; SRASBM, Self-Report of Aggression and Social Behavior Measure [20].

**Table 5 brainsci-11-01412-t005:** Findings relating to proactive aggression event-related potentials, according to year of publication.

Study	Year	*n*	Age	Sex	Region	PA Instrument	Wave	Correlation Coefficient/ Effect Size	Findings
Barratt et al. [95]	1997	101	26	M	North America	Offending record and semi-structured interview	P3	*na*	P3 amplitudes differed between impulsive and non-impulsive offender groups, but the groups did not differ in clinically-rated impulsivity.
Stanford et al. [96]	2003	28	33	M/F	North America	Offendingrecord	P3	*na*	P3 amplitudes for the PA group took marginally longer to peak in response to auditory stimuli.
Chen et al. [97]	2015	24	30	M	Asia	Offending record	P3	***d =* 1.33**	P3 amplitudes in the PA group showed less interference to sad cues.
Helfritz-Sinville and Stanford [98]	2015	58	19	M	North America	LHAQ IPAS	P3	*na*	P3 amplitudes showed less efficient processing of threat cues for the PA group.
Chen et al. [99]	2019	38	17	M	Asia	RPQ TAP	N2	***r* = 0.52** *r* = 0.12	PA was associated with reduced N2 wave during the decision phase of the TAP.

Statistical findings shown in bold are significant according to *p*-values < 0.05; *r*, Pearson correlation coefficient; *d*, Cohen’s d measure of effect size; *na*, not applicable; LHAQ, Lifetime History of Aggression Questionnaire [100]; IPAS, Impulsive/Premeditated Aggression Scales [101]; TAP, Taylor Aggression Paradigm [26].

**Table 6 brainsci-11-01412-t006:** Proactive aggression neuroimaging findings according to year of publication.

Study	Year	*n*	Age	Sex	Region	PA Instrument	Brain Region/ Structure	Modality	Findings
Raine et al. [102]	1998	65	33	M	North America	Offending record	Prefrontal, subcortical	PET	Predatory and affective murderers showed increased right subcortical glucose metabolism, compared to the non-offender control group.
Bobes et al. [103]	2013	54	29	M	North America	RPQ	Amygdala	sMRI fMRI	PA was negatively associated with right amygdala volume.
White et al. [104]	2013	59	14	M/F	North America	TRI	CSP	sMRI	PA was unrelated to CSP size.
Lozier et al. [105]	2014	46	14	M/F	North America	RPQ	Amygdala	fMRI	Right amygdala responses to fearful expressions negatively predicted PA in youths with high callous and unemotional traits.
Pardini et al. [106]	2014	56	26	M	North America	RPQ	Amygdala	sMRI	Low amygdala volumes were longitudinally associated with the development of PA starting in childhood.
Kuroki et al. [12]	2017	57	40	M	Asia	Offending record	Insula, planum polare, precueus	sMRI	Lower brain volumes were linked to a history of premediated violent acts among males with schizophrenia. This association may be related to psychotic symptomatology and not PA.
Yang et al. [107]	2017	106	14	M/F	North America	RPQ	ACC	sMRI	Left ACC volume was negatively associated with PA.
Farah et al. [108]	2018	156	35	M	North America	RPQ	Amygdala, ACC, insula	sMRI	PA was positively associated with right amygdala and left ACC volumes but unrelated to insular volume.
Kolla et al. [13]	2018	40	35	M	North America	RPQ	Precuneus, angular gyrus	fMRI	PA was negatively correlated with connectivity between the precuneus and angular gyrus among *MAOA-L* subjects. Both regions are nodes of the default mode network.
Craig et al. [109]	2019	233	13	M/F	Europe	RPQ	Left dorsal striatum	MRS	Glutamate concentration in the left dorsal striatum was inversely related to PA.
Siep et al. [110]	2019	36	36	M	Europe	RPQ	MPFC, STS	fMRI	PA was positively associated with connectivity in the MPFC and negatively with BOLD activity of the STS.
Zhu et al. [111]	2019	155	20	M/F	Asia	RPQ	Bilateral DLPFC, PCC, bilateral IPL, MPFC/ACC, precuneus	sMRI fMRI	PA was positively linked to GMD in bilateral DLPFC and negatively to GMD in the PCC. PA was negatively related to DLPFC and bilateral IPL connectivity as well as connectivity between PCC and other structures, such as MPFC/ACC, bilateral IPL, and precuneus.
Naaijen et al. [112]	2020	254	13	M/F	Europe	RPQ	Insula	sMRI	PA was negatively associated with amygdala volume.
Werhahn et al. [113]	2020	207	13	M/F	Europe	RPQ	Amygdala, precuneus	fMRI	PA was positively related to connectivity between the amygdala and precuneus.

RPQ, Reactive–Proactive Aggression Questionnaire [11]; TRI, Teacher Report Instrument [4]; CSP, cavum septum pellucidum; ACC, anterior cingulate cortex; MPFC, medial prefrontal cortex; STS, superior temporal sulcus; BOLD, blood oxygenation level-dependent; DLPFC, dorsolateral prefrontal cortex; PCC, posterior cingulate cortex; IPL, inferior parietal lobe; GMD, grey matter density.

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
