# Peer review of "Cold-Blooded and on Purpose: A Review of the Biology of Proactive Aggression"

_brainsci, 2021, doi:10.3390/brainsci11111412_

Round 1

Reviewer 1 Report

This review paper on the biological correlates of proactive aggression constitutes a very welcome addition to the literature on aggression, which is mostly focused on reactive aggression. I appreciate the critical reflection on the different methods to assess PA; the extensive number of biological correlates that were reviewed; and the clear writing style. The authors succeeded well in making this a quite brief and straightforward review, in spite of the large body of literature they summarize. The main message is clearly presented. The comments I have are mostly aimed to optimize parallel structures across the different result sections, and the improve a consistent use of terms, as well as accurate reference to the tables.

These are my specific comments:

Line 39: I do not think it is correct to refer to PA as `psychopathic in nature`, as it will elicit many different associations with the reader. It could be replaced by a statement evidencing the link between PA and psychopathic traits, as shown in empirical studies. In fact, this is already done some sentences later, so I think removing the statement of PA being psychopathic in nature is the better choice.

Line 112: it is argued that the methodology is based on the assumption that past behavior predicts future behavior. It would be good to add some references that actually showed this is the case, particularly for PA.

Line 114: `4. Psychophysiology`: This come quite sudden, as it not even introduced as the first part of the result section. What is needed is a general announcement of the overall structure of the paper and the findings early on in the paper. Relatedly, Table 1 is not generally introduced, and some sentence like `Table 1 reflects all findings on xx` is missing. In fact, table 1 is only referred to in the ``studies of children` section, but not in the sections on adolescents and adult, while these are also included in Table 1. The same goes for most of the other tables.

The title of Table 2 should include the term `hormones`.

Section 4 on psychophysiology is divided into sections on children, adolescents, and adults. For section 5 on hormones, no such subtitles are used, and the text instead is organized around typically developing children; at-risk/clinical samples; and hormonal reactivity. I think it would be much better for the paper if one and the same structure comes back in the different subsections.

In the discussion, I am missing a reflection on the fact that PA is actually a multi-facetted concept, including e.g. goal orientation and a lack of provocation. Many of the studies use measures or proxies of PA that in fact do not adequately assess all these components. This was partly addressed in the method section part 3, but requires re-evaluation after all findings have been presented. The discussion would profit from adding a critical reflecting on this, as well as the possible implications of this.

Reviewer 2 Report

In this manuscript, the authors conducted a systematic review of studies on the biology of proactive aggression, including findings on cortisol and other hormones, molecular genetic, neurophysiology, and neuroimaging. This topic is timely and includes many different biological approaches that have been employed to investigating the basis of proactive aggression.  It provides an important summary of the field. The manuscript is also generally well-written. However, below are several suggestions that may greatly improve the manuscript.

Major comments:

1), for the reviewed findings in Table 1, 2, 4, 5, 6, rather than simply significant or nonsignificant, the authors should provide the original correlation coefficients or effect size measures for all studies. This is an important trend of most, if not all, scientific fields. 

2), a schematic illustration incorporating the reviewed mechanisms or biological associates may greatly enhance the manuscript. An image of the brain that summarizes the neuroimaging studies or showing important regions indicated by the reviewed studies is also preferred. 

Minor comments:

1), abstract, line 15: a robust association between?

2), line 75: what is "an experimental, survey-based measure"?

Round 2

Reviewer 2 Report

Thank the authors for addressing my concerns.

Author Response

Thank you again for suggestions.